# Eye-Gaze direction triggers a more specific attentional orienting compared to arrows

**Jeanette A. Chacón-Candia**[1,2]*, **Juan Lupiáñez**[1], **Maria Casagrande**[3], **Andrea Marotta**[1]

**1** Department of Experimental Psychology, and Mind, Brain, and Behavior Research Center (CIMCYC), University of Granada, Granada, Spain, **2** Dipartimento di Psicologia, Sapienza Università di Roma, Rome, Italy, **3** Dipartimento di Psicologia Dinamica e Clinica, Sapienza Università di Roma, Rome, Italy

* chaconcandia@ugr.es

**Data Availability Statement:** All data for the conduction of this study are publicly available via

## Abstract

Numerous studies have shown that eye-gaze and arrows automatically shift visuospatial attention. Nonetheless, it remains unclear whether the attentional shifts triggered by these two types of stimuli differ in some important aspects. It has been suggested that an important difference may reside in how people select objects in response to these two types of cues, eye-gaze eliciting a more specific attentional orienting than arrows. To assess this hypothesis, we examined whether the allocation of the attentional orienting triggered by eye-gaze and arrows is modulated by the presence and the distribution of reference objects (i.e., placeholders) on the scene. Following central cues, targets were presented either in an empty visual field or within one of six placeholders on each trial. In Experiment 2, placeholder-objects were grouped following the gestalt's law of proximity, whereas in Experiment 1, they were not perceptually grouped. Results showed that cueing one of the grouped placeholders spreads attention across the whole group of placeholder-objects when arrow cues were used, while it restricted attention to the specific cued placeholder when eye-gaze cues were used. No differences between the two types of cues were observed when placeholder-objects were not grouped within the cued hemifield, or no placeholders were displayed on the scene. These findings are consistent with the idea that socially relevant gaze cues encourage a more specific attentional orienting than arrow cues and provide new insight into the boundary conditions necessary to observe this dissociation.

## Introduction

The capacity to follow the focus of attention of another individual is of great importance for the development of social communication [1, 2].

In order to understand what others are paying attention to, we usually rely on information provided through non-verbal communication, such as gestures, postures, and the direction of the gaze [3]. The perception, interpretation and evaluation of the information obtained through these sources help us inquire about other people's intentions and mental states and, consequently, anticipate their next step and increase the probability of successfully building

Open Science Framework and can be accessed at
https://osf.io/xmq9v.

**Funding:** This work was supported by a research
project (PID2020-114790GB-I00) by the Spanish
Ministry of Science and Innovation/AEI (https://
www.ciencia.gob.es/) to JL, a research project (B-
SEJ-572-UGR20) by the Regional Government of
Andalusia (https://www.juntadeandalucia.es/) to
AM and a Ph.D. fellowship in Psychology and
Cognitive Science by "La Sapienza" The University
of Rome (https://www.uniroma1.it) to JACH-C. The
funders had no role in study design, data collection
and analysis, decision to publish, or preparation of
the manuscript.

**Competing interests:** The authors have declared
that no competing interests exist.

social interactions [4–6]. Together with other biologically relevant stimuli [3, 7] averted gaze of another person can shift the observer's attention in the same direction as the observed gaze (e.g., [8, 9], see [10] for review), allowing the establishment of "joint attention" [11]. This behaviour has been considered highly beneficial to individuals and has been a crucial step in the development of social-communicative skills [2, 12, 13]. For this reason, many studies have investigated the mechanisms underlying this phenomenon.

Friesen and Kingstone [14] were the first to demonstrate that looking at eye-gaze will trigger the shift of our attentional focus into the gazed-at location. They used a variant of the classic visuospatial cueing paradigm [15] in which, at the centre of the screen, a schematic face appeared, gazing either straight ahead, left or right. The participants' task was to detect, locate or discriminate a target that would appear congruently at the gazed location or incongruently at the opposite one. They found that targets appearing at the congruent location were detected, located, or discriminated more quickly than targets appearing at the incongruent one. Since then, an increasing number of researchers have further studied this effect using the same or slight variations of this cueing paradigm. Results repeatedly demonstrated that even when gaze direction is not predictive of target location (e.g., [16–19]) or is counterpredictive (e.g., [8]), the gaze shift automatically directs the observer's attention to the same location indicated by it (see [6, 10] for a review).

Based on these behavioural findings and the evolutionary and social significance of eye gaze [5], several authors have suggested that the attentional orienting triggered by the eye-gaze direction may represent a unique attentional process that can be differentiated from that produced by directional stimuli with no biological relevance, such as arrows (e.g. [9, 18, 20]), which have proven as well to facilitate attentional orienting, even if they are non-predictive [21]. In this regard, many studies have tried to answer whether arrow and gaze cues produce the same or different behavioural or neural effects, leading to mixed results, with some of them finding a significant difference between the two stimuli and others suggesting that the effect triggered by them is indistinguishable (e.g., [22–26]).

However, clarifying this debate, recent meta-analytical evidence [27] has shown no behavioural differences between the attentional orienting triggered by eye-gaze and arrow cues. For instance, it remains unclear whether the attentional shifts induced by these two types of cues differ in some other important aspects. Recently, a study by Marotta, Lupiáñez, Martella, and Casagrande [18] suggested that the source of a possible difference between eye-gaze and arrow attentional cues may lie in the dissimilar way people select objects in response to these two types of cues. In particular, they speculated that "biologically and socially relevant gaze cues may encourage more specific attentional orienting, compared to arrow cues, since a specific intention may be automatically attributed to gaze and not to arrows" ([18], p. 333). Consistent with this view, they found that when using eye-gaze as a cue, attention is directed specifically to the location or part of the object being looked at. In contrast, when using an arrow, attention spreads across the entire cued object.

The property of gaze cues to induce "specific" attentional orienting has also been corroborated by Wiese, Zwickel, and Müller [28], showing that when previewed location placeholders were used, gaze cues induced a facilitation effect only when targets appeared inside the exact placeholder pointed at, but not when targets appeared in different spatially located objects within the cued hemifield. However, when no placeholders were presented, gaze cueing effects were detectable in response to the specific cued location but also spread across the entire cued hemifield. In light of these findings, another person's gaze may trigger a specific attentional orienting only when an object is presented in the visual scene.

Considering the importance of orienting attention to the same object of others' attentional direction to establish a social joint attention episode, this makes perfect sense. In other words,

another person's gaze may induce a specific attentional orienting only when an object is present in the environment and can be interpreted as the goal of the gaze. However, this should not be observed in response to arrow cues since arrows have a directional property, like gaze, but no biological or social significance. However, to date, no studies have directly compared the attentional selection produced by these two types of stimuli in the presence or absence of placeholders within the visual field. To accomplish this aim, in the present study, we have used a paradigm very similar to that used by Wiese and colleagues [28], in which, in response to gaze and arrow cues, participants had to respond to targets presented in one of three possible locations within a cued hemifield: 0° and +/-60° from the horizontal meridian. Placeholder objects for the targets will be presented on half the trials (placeholder-present condition), while on the other half, no placeholders will be presented (placeholder-absent condition).

In the placeholder-present condition, we expected that gaze cues would elicit a specific attentional orienting benefit only for targets presented within the object (i.e., placeholder) looked at, but not for targets appearing in different spatial locations within the cued hemifield. Arrows should elicit a more general attentional benefit across the entire cued hemifield. As mentioned above, the cued object should be interpreted as the goal of another person's attention only in response to gaze cues (i.e., looked at object) but not in response to arrows. On the other hand, no difference between gaze and arrow attentional effect should be observed when no objects are presented on the scene (placeholder-absent condition), cueing effects spreading across the entire cued hemifield with both gaze and arrow cues.

## Experiment 1

### Method

**Participants.** Thirty-seven undergraduate students (24 female; mean age: 22 years) gave their informed consent before voluntarily participating in this research. There was no clear experiment of reference for computing the needed sample size in our first experiment, as this was the first time our paradigm was used. We could use as reference the study by Wiese et al., [28], but they did not compare arrows and gaze, which was critical for our experiment. Instead, we could use Marotta et al. [18] experiments, in which objects instead of group of objects (i.e., placeholders) were used, but they did compare gaze with arrow cues. Marotta et al. [18] used samples of 24 and 30 participants, so we decided to use a minimum of 36 participants for Experiment 1. Because the sample size was not computed a priori based on the effect size of a previous study, we used G*Power [29] to compute sensitivity of our specific relevant analyses regarding the orienting effects (t-test). With our sample size (37 participants) the minimum effect size that could be detected for $\alpha = 0.5$, and $1-\beta = 0.80$, is Cohen's $dz = 0.417$, which is higher than most effect sizes of interest.

All participants had normal or corrected to normal vision and were unaware of the purpose of the experiment. In this and the following experiments, participants received course credits for their participation. All experiments were approved by the Ethical Committee of the University of Granada (175/CEIH/2017) and conducted in conformity with the ethical standards of the Declaration of Helsinki.

**Apparatus and stimuli.** The cueing-discrimination task used in this experiment was presented on a 21-inch VGA colour monitor of a computer running E-Prime software [30] to control the presentation of the stimuli, timing operations, and data collection.

On the hemifield placeholder-present condition, the fixation display consisted of three placeholder boxes presented within each hemifield at 0° and +/-60° from the horizontal meridian; the central fixation stimuli changed depending on the cue type. For the arrow trials, a horizontal line was presented at the centre of the screen, and for the gaze trials, the display was a

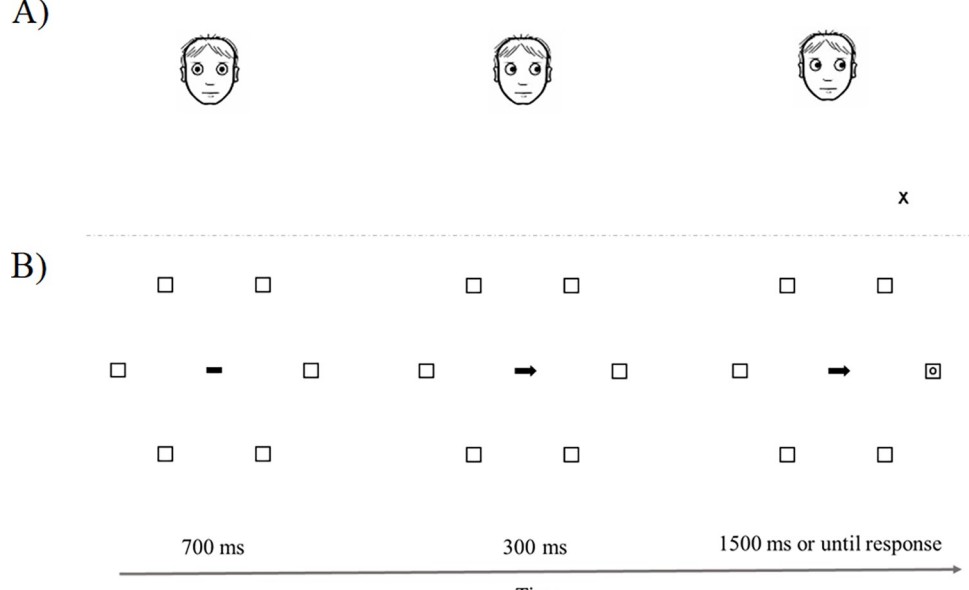

**Fig 1. Schematic view of a trial sequence for both the gaze cue and the arrow cue conditions.** The example represents: A) gaze-cue/placeholder-absent/same-hemifield condition, and B) arrow-cue/placeholder-present/same-location/same-hemifield condition.

schematic face with the eyes looking straight. During experimental trials, the face pupils, or the appearance of an arrowhead, signalled left or right from fixation. Target stimuli were the letters "X" or "O". The background of the screen was white, and all the stimuli were black.

**Procedure.** After giving their informed consent, participants were seated at about 55cm from a computer screen in a quiet, dimly lit room. Trials started with a fixation display that differed depending on the cue type. In gaze cueing trials, a schematic face with a straight gaze was presented as fixation, whereas, in arrow cueing trials, the fixation stimulus was a horizontal line centred on the screen. This display was presented for 700ms; then, a change was made to the arrow or eye gaze fixation points to indicate left or right on the horizontal meridian (importantly, no other position or placeholder was directly cued). Following the presentation of the cue, a target (either the letter "X" or "O") appeared unpredictably in one of six possible locations (see Fig 1).

Stimulus onset asynchrony (SOA) was 300ms. Cue and target remained on the screen until a response was given or for 1500ms in case of no response. Then, a blank display was presented for 700ms. Targets appeared either in one of the three placeholder boxes presented within each hemifield (placeholder-present condition) or at one of the same positions in an empty space when no placeholder boxes were presented (placeholder-absent condition).

Participants were required to discriminate the letter "X" or "O" by pressing either the "M" key (with the right hand) or the "Z" key (with the left hand) on the computer keyboard, depending on the target letter that was presented. Half of the participants pressed "M" for target "X" and "Z" for target "O", whereas the other half received the reversed mapping. They were also instructed to respond as quickly and accurately as possible and maintain central fixation throughout all trials. They were informed that the direction of the central stimuli did not predict the location of the target, so they should ignore it.

Cue direction, target stimuli, target location, and placeholder presence were randomly interspersed within each block of trials, whereas cue type was manipulated between blocks in a

Same-Location/Same-Hemifield

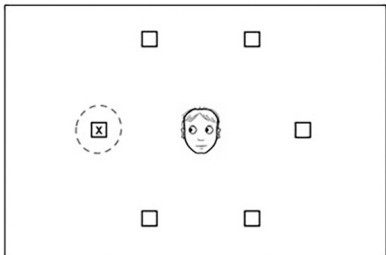

Opposite-Location/Opposite-Hemifield

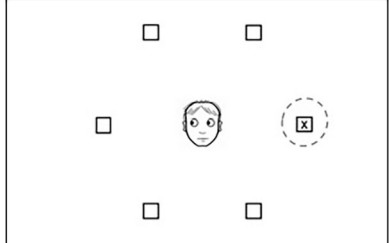

Same-Hemifield

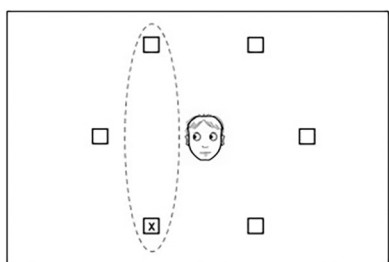

Opposite-Hemifield

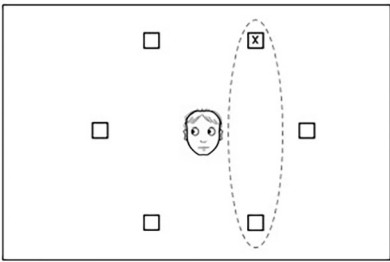

**Fig 2. Illustration of the four types of cue-target relation of Experiment 1.** The images represent the gaze-cue in a placeholder-present condition. The cue-target relation for the placeholder-absent condition was the same, with the exception that no placeholder boxes were presented on the scene.

counterbalanced order. There were two experimental blocks of 288 trials each (one for each cuetype), each preceded by a practice block of eight trials (where participants received feedback for their performance), summing up 592 trials in total.

**Design.** Three-factor repeated measure design was used to analyse an overall effect in this experiment, 2 (cue-type) x 2 (placeholder-condition) x 4 (validity). The cue-type had two levels, arrow and eye-gaze; placeholder-condition consisted of placeholder-present and placeholder-absent conditions, and the four validity levels were same-location/same-hemifield, opposite-location/opposite-hemifield, same-hemifield and opposite-hemifield trials.

Given our main interest on specific attentional orienting mechanisms, t-test analyses were performed to analyse specific effects of validity (general-cueing and hemifield-effects), and its modulation by relevant variables. For the general-cueing effect, the comparison of cue-target relations consisted of same-location/same-hemifield trials vs opposite-location/opposite-hemifield trials; for the hemifield-effect, the cue-target relation consisted of same-hemifield vs opposite-hemifield trials (see Fig 2).

## Results

For the reaction time analysis, trials with correct responses faster than 100ms or slower than 1200ms (0.5%), and incorrect response trials (5.69%) were excluded. Mean RT, standard deviations, and error percentage for all conditions are shown in Table 1.

A cue-type (arrows vs. gaze) x placeholder-condition (placeholder-present vs. placeholder-absent) x validity (same-location/same-hemifield, opposite-location/opposite-hemifield, same-hemifield and opposite-hemifield) repeated measures ANOVA was performed to analyse an overall effect.

**Table 1. Mean reaction times (RT), standard deviation (SD), and percentage of incorrect responses (%IR) as a function of the placeholder-condition, type of cue, and cue-target (CT) relation in Experiment 1.**

| | Placeholder-Present Condition | | | | | | Placeholder-Absent-Condition | | | | | |
| --- | --- | --- | --- | --- | --- | --- | --- | --- | --- | --- | --- | --- |
| | Arrow | | | Gaze | | | Arrow | | | Gaze | | |
| CT relation | RT | *SD* | %IR | RT | *SD* | %IR | RT | *SD* | %IR | RT | *SD* | %IR |
| **Same-Location/ Same-Hemifield** | 486 | 69.02 | 5.89 | 486 | 61.11 | 6.70 | 487 | 67.06 | 5.97 | 471 | 57.97 | 4.38 |
| **Opposite-Location/ Opposite-Hemifield** | 507 | 77.57 | 7.03 | 493 | 59.54 | 5.73 | 495 | 72.09 | 6.68 | 486 | 60.05 | 4.95 |
| **Same-Hemifield** | 514 | 70.52 | 6.07 | 506 | 55.6 | 5.65 | 493 | 64.92 | 5.22 | 482 | 58.93 | 4.91 |
| **Opposite-Hemifield** | 513 | 70.29 | 7.30 | 512 | 62.79 | 5.88 | 492 | 64.76 | 4.88 | 493 | 62.55 | 5.31 |

The analysis reported a main effect of placeholder-condition ($F_{1,36}$ = 21.33, p = < .001, $\eta^2_p$ = 0.372), showing that overall reaction times were faster when no placeholders were presented on the scene ($M$ = 487, SD = 63.36) than when placeholders were presented ($M$ = 502, SD = 66.28). A main effect of validity was also found ($F_{3,108}$ = 30.24, p = < .001, $\eta^2_p$ = 0.457), showing that reaction times were faster when the target appeared at the same-location/same-hemifield ($M$ = 483, SD = 63.65), followed by opposite-location/opposite-hemifield ($M$ = 495, SD = 67.51), same-hemifield ($M$ = 499, SD = 63.26) and opposite-hemifield ($M$ = 502, SD = 65.27) respectively.

The placeholder-condition X validity interaction was also significant ($F_{3,108}$ = 6.20, p = < .001, $\eta^2_p$ = 0.147). Partial ANOVAs for each validity condition showed that when the targets appeared at the same-location/same-hemifield, there were no differences related to the presence or absence of placeholders in the scene (p>.05). However, when targets appeared at opposite-location/opposite-hemifield, same-hemifield and opposite-hemifield participants were significantly faster when no placeholder objects were presented (all $ps$ < .05).

More importantly, separate t-test analyses were conducted to analyse, on the one hand, the general-cueing effect (same-location/same-hemifield vs. opposite-location/opposite-hemifield) and, on the other, the hemifield-effect (same-hemifield vs opposite-hemifield), in both the placeholder-absent and the placeholder-present conditions. The results revealed that the general-cueing effect was significant for both the placeholder absent ($t(36)$ = -3.889, p = < .001, $d$ = -0.639) and the placeholder-present conditions ($t(36)$ = -4.719, p = < .001, $d$ = -0.776), showing that in general, reaction times were faster when targets appeared at the same-location/same-hemifield trials than at the opposite-location/opposite-hemifield trials regardless the presence of placeholders on the scene (see Fig 3). When analysing the hemifield-effect, no significant effect was found for any of the placeholder conditions (all ps>.05).

Importantly, neither the main effect of cue-type, nor its interaction with any other variable reached significance (all $ps$ >.05).

## Discussion

This experiment tested whether eye-gaze attentional cues trigger more specific attentional orienting than arrows when placeholder objects are presented on the signalled hemifield. However, the results of this experiment showed that arrows and eyes triggered very similar attentional cueing effects in both placeholder-absent and present conditions. In particular, with both cues, a significant attentional benefit was only observed for targets appearing at the specifically cued location but not for targets appearing in different spatial locations within the cued hemifield.

At first sight, these findings seem to suggest that attention triggered by social and non-social cues is not modulated by the presence of placeholders on the scene, and they are consistent with the literature, which has generally reported similar behavioural cueing effects for gaze and arrows in the normotypical population (for review, see [32]). On the other hand, they

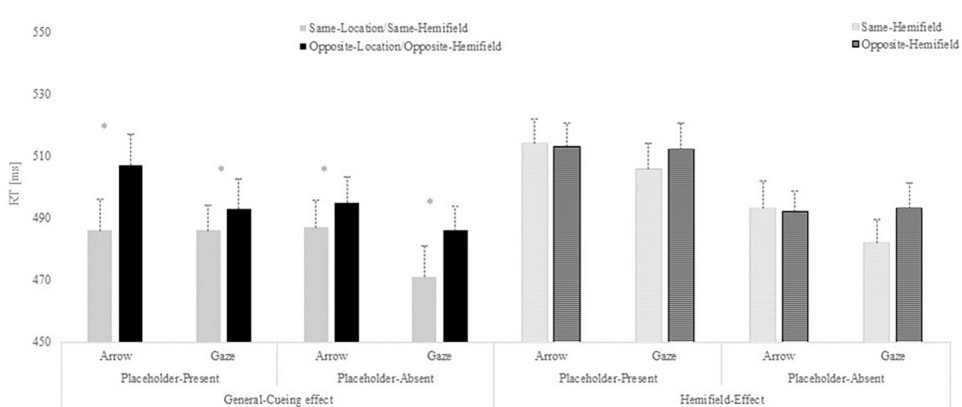

**Fig 3. Reaction times (RTs) results from Experiment 1.** Results are shown separately for the general-cueing effect (Same-Location/Same Hemifield vs. Opposite-Location/Opposite Hemifield) and the hemifield-effect (Same-Hemifield vs. Opposite Hemifield). Mean RTs presented for each type of cue as a function of the cue-target relation in the placeholder-present and placeholder-absent conditions. Error bars represent the standard error of the mean, computed following Cousineau's [31] method to eliminate variability between participants.

seem to contrast with our hypothesis according to which attentional benefits should be observed only for targets presented in the specific object (or part of an object) when signalled by eye-gaze cues, and for all the targets, independently from their position in the cued hemifield, when signalled by arrows.

Indeed, we assumed that arrows should elicit a more general attentional benefit spreading across the cued hemifield, based on our previous findings showing that arrows, but not eye-gaze, allow attentional shifts to spread through to the entire surface of an object presented in the cued visual field. Nevertheless, given the specific paradigm we used in our previous experiment, an alternative explanation could be plausible. As shown in Fig 2, the six objects were equidistant and distributed across the circle of objects that served as a background fixation display. Then it makes sense that only a general-cueing effect is observed for both arrows and gaze. It could then be possible that arrows trigger attentional orienting spreading the cued object's entire surface but not across the entire cued hemifield. This would explain why in the present experiment, an attentional effect was observed only for targets appearing at the specifically cued location or object, as both arrows and gaze similarly orient attention to the specifically cued signalled object. In the following experiments, we decided to modify the proximity between the objects within each hemifield so that participants would perceive one easily segregated group of objects.

## Experiment 2

The goal of experiment 2 was to investigate whether, by manipulating the distribution of placeholders within the hemifield (i.e., following the gestalt's law of proximity [33, 34]), cues would trigger attention not only to the specific cued object but also towards the entire group of signalled placeholder objects. In particular, since there is evidence that the attention system similarly treats perceptually grouped objects [35, 36], we expected that by grouping by proximity the placeholders, the attentional effect would be similar to the one found by Marotta and colleagues [18]: attention would spread to the whole group of placeholder objects only when using an arrow, while, when using eye-gaze, attention would be directed just to the specific cued placeholder. Moreover, we did not expect such an effect when no placeholders were presented on the scene.

## Method

**Participants.** A new sample of seventy-five undergraduate volunteers (64 females; 18–35 years) were recruited through an experimental online platform from the University of Granada. Participants followed the protocol equally and had the same characteristics as those in experiment 1. Given the online collection of data we decided to double the sample size.

Again, as in experiment 1, because the sample size was not computed a priori based on the effect size of the previous study, we used G\*Power [29] to compute sensitivity of our specific relevant analyses regarding the orienting effects (t-test). With our sample size (75 participants) the minimum effect size that could be detected for $\alpha = 0.5$, and $1-\beta = 0.80$, is Cohen's $dz = 0.290$, which is higher than most effect sizes of interest.

**Apparatus and stimuli.** Unlike Experiment 1, the cueing discrimination task was designed using the graphical experiment builder OpenSesame [37]. As shown in Fig 4, the stimuli in this experiment were nearly the same as those used in the previous experiment, except for the placeholder boxes distribution. This time in the displays of the placeholder-present condition, the three placeholder boxes subtending within each hemifield were located at 0˚, +/- 45˚ and +/-90˚ from the horizontal meridian and were randomly presented in two possible distributions (+/-45˚ from the vertical meridian). No other changes were made to the stimuli.

**Procedure.** Participants completed this experiment online. They were provided with a link to a survey (using the Lime Survey platform; https://www.limesurvey.org/) to complete the informed consent, receive instructions and be redirected to the online behavioural task (hosted on a JATOS server). All participants were given a clear indication of using a computer to complete the task in order to prevent them from utilising a different device (e.g., smartphone or tablet). Furthermore, the experiment was programmed in a way that a keyboard was needed for the correct recording of responses.

Furthermore, the procedure of this experiment was similar to the one used in experiment 1, although some changes were made. First, the presentation of the placeholder-condition (present and absent) was separated into two blocks. Second, the order of spatial cues (arrow and gaze) was randomly interspersed within each block of trials. Third, as stated above, in the placeholder-present condition, the positions of the six placeholder boxes were grouped into

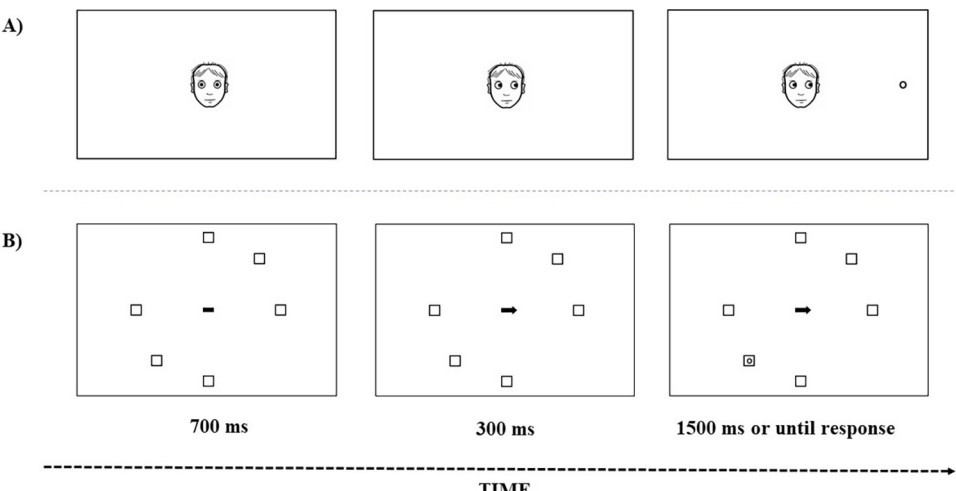

**Fig 4. Schematic view of a trial sequence for both the gaze cue and the arrow cue conditions of Experiment 2.** The example represents: A) gaze/placeholder-absent/same-location/same-group condition, B) arrow/placeholder-present/opposite-group condition.

Same-Location/Same-Group

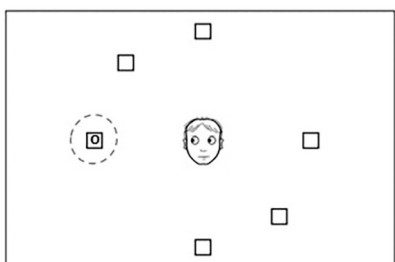

Opposite-Location/Opposite-Group

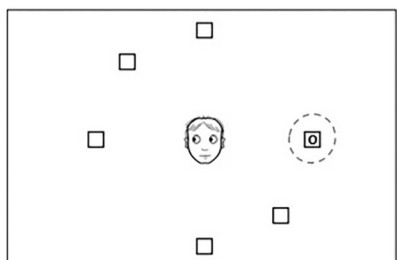

Same-Group

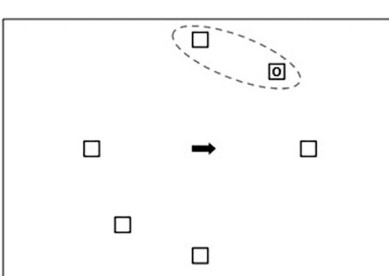

Opposite-Group

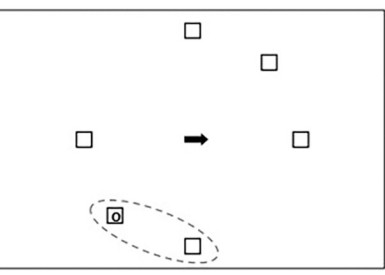

**Fig 5. Illustration of the four types of cue-target relation of Experiments 2.** The placeholder-group tilted orientation shown here is -45˚ from vertical. The top images represent an example of gaze cue in a placeholder-present condition; the bottom images represent the arrow cue in a placeholder-present condition. The cue-target relation for the placeholder-absent condition was the same, with the exception that no placeholder boxes were presented on the scene.

quadrants, appearing at radial distances of 0˚, +/- 45˚ and +/- 90˚ from the horizontal axis of a central stimulus (see, Fig 4) and were randomly positioned in two possible distributions (+/-45˚ from the vertical meridian). Finally, the six possible target positions were adapted as the distribution of the placeholder boxes described above (0˚, +/- 45˚ and +/- 90˚ from the horizontal axis) for both placeholder-present and placeholder-absent conditions. The four critical cue-target relations for the analysis were almost equal to the previous experiment but just adapted to the new possible target positions (see, Fig 5). The remaining characteristics of the procedure were the same as in experiment 1.

**Design.** As in experiment 1, in this experiment, an overall effect was analysed by using a three-factor repeated measure design, 2 (cue-type) x 2 (placeholder-condition) x 4 (validity). Similar to experiment 1, the cue-type had two levels, arrow and eye-gaze; placeholder-condition had two levels, placeholder-present and placeholder-absent, and validity had four levels, now-called same-location/same-group, opposite-location/opposite-group, same-group and opposite-group. To analyse the general-cueing effect and the now called grouping-effect (targets appearing at +/- 45˚ and, +/- 90˚ from the horizontal meridian of the cue), T-test analyses were performed separately for each placeholder condition. For the general-cueing effect, the comparison of cue-target relations consisted of same-location/same-group trials vs opposite-location/opposite-group trials; for the grouping-effect, the cue-target relation consisted of same-group vs opposite-group trials. When no placeholders were presented, the cue-target relations corresponding to same-group and opposite-group conditions were created by distributing the up and down trials between those two types of cue-target relations. The order of blocks of each placeholder condition (present/absent) was counterbalanced across participants.

**Table 2. Mean reaction times (RT), standard deviation (SD), and percentage of incorrect responses (%IR) as a function of placeholder-condition, type of cue, and cue-target (CT) relation in Experiment 2.**

| | Placeholder-Present Condition | | | | | | Placeholder-Absent-Condition | | | | | |
|---|---|---|---|---|---|---|---|---|---|---|---|---|
| | Arrow | | | Gaze | | | Arrow | | | Gaze | | |
| CT relation | RT | SD | %IR | RT | SD | %IR | RT | SD | %IR | RT | SD | %IR |
| Same-Location/Same-Group | 519 | 76 | 6.00 | 536 | 77.65 | 6.73 | 520 | 69.33 | 5.81 | 516 | 66.47 | 6.14 |
| Opposite-Location/Opposite-Group | 553 | 81.69 | 8.78 | 547 | 77.32 | 7.97 | 529 | 74.38 | 5.25 | 529 | 68.18 | 6.86 |
| Same-Group | 559 | 80.34 | 6.00 | 563 | 78.59 | 6.58 | 529 | 65.22 | 5.53 | 532 | 70.07 | 5.37 |
| Opposite-Group | 571 | 82.13 | 6.92 | 565 | 75.63 | 6.92 | 529 | 65.43 | 5.60 | 535 | 70.72 | 5.71 |

## Results

Correct response trials with RT faster than 100ms or slower than 1200ms (0.8%) and incorrect response trials (6.29%) were excluded from the RT analysis. Mean RT, standard deviations, and error percentage for all conditions are shown in Table 2.

A cue-type (arrows vs. gaze) x placeholder-condition (placeholder-present vs. placeholder-absent) x validity (same-location/same-group, opposite-location/opposite-group, same-group and opposite-group) repeated measures ANOVA was performed to analyse an overall effect.

The analysis reported a main effect of placeholder-condition ($F_{1,74} = 26.86$, $p = < .001$, $\eta^2_p = 0.266$), showing that overall reaction times were faster when no placeholders were presented on the scene ($M = 527$, SD = 68.63) than when placeholders were presented ($M = 552$, SD = 79.88). A main effect of validity was also found ($F_{3,222} = 61.25$, $p = < .001$, $\eta^2_p = 0.087$), showing that reaction times were faster when the target appeared at the same-location/same-group ($M = 523$ $SD = 72.56$), followed by opposite-location/opposite-group ($M = 540$, SD = 75.96), same-group ($M = 546$, SD = 75.04) and opposite-group ($M = 550$, SD = 75.59) respectively.

The placeholder x validity interaction was also significant ($F_{3,222} = 18.07$ $p = < .001$, $\eta^2_p = 0.196$). Partial ANOVAs showed that when the targets appeared at the same-location/same-group, no differences related to the presence or absence of placeholders in the scene were found (p>.05); nonetheless, when targets appeared at the opposite-location/opposite-group, same-group, and opposite-group, participants were significantly faster when no placeholder objects were presented (all $p$s < .001).

The main effect of cue-type, nor its interaction with the variables placeholder condition or validity reached significance (all $p$s >.05). Nonetheless, a three-way interaction of placeholders x cue type x validity was found ($F_{3,222} = 7.11$ $p = < .001$, $\eta^2_p = 0.088$), showing that when no placeholders were presented, as expected, only the main effect of validity was significant ($F_{3,222} = 9.75$, $p = < .001$, $\eta^2_p = 0.016$), whereas when placeholders were presented both the main effect of validity ($F_{3,222} = 72.99$, $p = < .001$, $\eta^2_p = 0.496$), and the cue type x validity interaction ($F_{3,222} = 6.73$, $p = < .001$, $\eta^2_p = 0.083$), were significant.

Indeed, T-test analyses, separately conducted for placeholder absent and present conditions and for each cue type, revealed that when placeholders were absent, it was possible to observe a general-cueing effect for both gaze ($t(74) = -2.376$, p = 0.02, $d = -0.274$) and arrows ($t(74) = -2.027$, p = 0.046, $d = -0.234$); in this condition, no grouping effect was found for any of the cue types (all $p$s>.05). When placeholders were presented on the scene, the general-cueing effect was also observed for both gaze ($t(74) = -2.472$, p = 0.016, $d = -0.285$) and arrows ($t(74) = -6.247$, p = < .001, $d = -0.721$). Nevertheless, and importantly, in this condition, the analysis revealed a main effect of grouping but this was observed only when arrows were used as cue ($t(74) = -3.618$, $p = < .001$, $d = -0.418$); when gaze was used as cue, no grouping-effect was observed ($t(74) = -0.769$, $p = .445$, $d = -0.089$; see Fig 6).

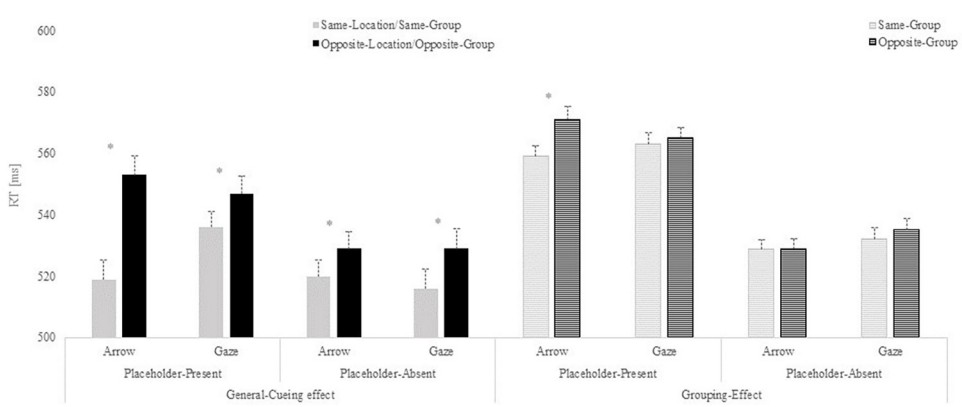

**Fig 6. Reaction times (RTs) results from Experiment 2.** Results are shown separately for the general-cueing effect and the grouping-effect. Mean RTs presented for each type of cue as a function of the cue-target relation in the placeholder-present and placeholder-absent conditions. Error bars represent the standard error of the mean, computed following Cousineau's [31] method to eliminate variability between participants.

## Discussion

As in the previous experiment, no facilitation effect was observed for any cue beyond the specifically cued location when no placeholder objects were presented. However, experiment 2 was conducted to assess whether attention would spread to an entire group of placeholders within a hemifield when using a central non-informative arrow cue and whether eye-gaze will trigger attention just to the specific location or placeholder of the group that is being signalled. Results showed that both arrow and gaze cues provoke attentional facilitation when targets appear at the exact object/location that is being pointed at (general-cueing effect). On the other hand, only arrows, but not eye-gaze, seemed to orient attention to targets appearing in the same group of objects but in a different position than the one indicated by the cue (grouping-effect/placeholder-present condition).

These findings can lead us to speculate that biologically relevant stimuli such as eye-gaze may trigger more specific attentional orienting than arrows due to the particular intention that we may attribute to the others' focus of attention. However, this specific gaze effect is only observed when measuring attentional facilitation beyond the specifically cued location/object, where a general-cueing effect is observed for arrows and gaze, consistently with the literature. Furthermore, in order for attention to spread to close objects, these must be perceptually organized into distinct groups of objects, as in this experiment, and differently from the previous one. Interestingly, attention spread to nearby objects within the group only with arrow cues even under these conditions. Conversely, when a gaze cue was used, attention was restricted to the specifically cued object within the group.

## General discussion

The present study aimed to explore through a series of two experiments whether the attentional orienting in response to non-predictive arrow and eye-gaze cues differs when placeholder objects are presented on the scene.

Results suggest that when several placeholders are grouped into a perceptual object as a function of Gestalt principles of proximity (Experiment 2), gaze and arrows cues elicit attentional effects similar to those first reported by Marotta and colleagues ([18], see [38] for replication). In particular, they showed that when objects were present in the display, eye-gaze cues directed attention to the specific part of the cued object, while arrow cues spread attention to

the entire signalled object. Here, we extend these results to new displays in which no entire objects but groups of placeholders, grouped according to their proximity, were presented. In particular, it was observed that attention spread to the whole group of placeholder objects only when using an arrow, while it was restricted to the specific cued placeholder when eye-gaze cues were used. This pattern of results has been replicated in a different on-line study with a larger sample size in which we investigated the modulation of gender/sex over the orienting effects observed with arrows and gaze [39].

On the other hand, when placeholder objects were not grouped within a cued hemifield, as in Experiment 1, arrows and eyes triggered very similar attentional cueing effects with a significant attentional benefit only for targets appearing at a specifically cued location or placeholder, but not for targets appearing in other spatial locations or placeholders. The fact that with arrow cues, the RT advantage for targets presented in the placeholders of the cued hemifield is not present when placeholders are not grouped suggests that arrows trigger attentional orienting spreading to the entire surface of a cued perceptual object but not across the entire cued hemifield, neither when different ungrouped objects are spread out in the hemifield (i.e., in the placeholders present condition in Experiment 1), nor in the absence of any object (in the placeholders absent condition in Experiments 1 and 2).

As a potential limitation, it is important to note that for the arrow cues, the horizontal line is present and then the arrowhead appears at cue onset, whereas for the gaze cues, the pupils are present and then move to the left or the right at cue onset. Also, eye movements were not controlled, which could allow for the differences observed between cue types. Note, however, that if these differences were due to these factors, they would be observed independently of the presence or absence of placeholders, and whether they could be easily grouped or not into two objects. However, the differences seem to be related to how the two cue types interact with attention to groups of objects.

Indeed, the present results have important implications for the perceptual grouping literature, as well as the social attention literature. Interestingly, the influence of Gestalt principles in attentional selection tasks had been previously established in earlier research using peripheral cues (e.g., [36, 40]). The offered results extend these findings to central non-predictive non-social cues. It has been previously suggested that cueing a portion of an object spreads attention across the entire object when arrow cues are used, while it restricts attention at the specific portion of the cued object when eye-gaze cues are used. The present results extend this notion, suggesting that this attentional dissociation is also observed when grouped objects are cued by eye-gaze and arrow cues. The boundary conditions for this effect seem to be related to Gestalt laws of perceptual grouping, as no grouping-effect was observed in Experiment 1 when distance and similarity perhaps led to the perceptual segregation of the display on a single group of objects (i.e., the six placeholders) rather than into two groups of objects (one cued and the other uncued) as in Experiment 2.

Therefore, both peripheral cueing and the effects of symbolic non-predictive non-social cues seem to be triggered automatically and mediated by object-based processes. Importantly, although social directional cues like gaze might produce an effect of a similar nature, as the common effect observed with the standard gaze cueing paradigm and the general-cueing effect observed in our experiments, they must produce an extra effect that restricts attention to the specifically looked-at location. The idea that gaze triggers both an effect similar to the one induced by non-social cues [27] and an extra specific effect has also been shown with other paradigms. Indeed, Marotta and colleagues [41], in a study in which both behavioural and electrophysiological data were collected, observed that arrows and gaze produce a similar effect at earlier event-related components (P1 and N1) but opposite effects at later components (N2 and P300).

Thereby, the present results seem to argue in favour of the idea that biologically relevant stimuli such as eye-gaze may trigger a unique attentional process, qualitatively distinct from the attentional process triggered by non-biologically relevant stimuli such as arrows. Marotta and colleagues [18] suggested that this specific attentional orienting effect of eye-gaze might be mediated by the automatic attribution of intention to gaze and not to arrows. This notion seems to be supported by the present study results and by the observations of Vuilleumier [42] and Wiese et al., [28], showing that when reference objects are presented on the scene, gaze cues trigger a facilitation effect but only to the specific gaze-at object.

For decades, an eye-gaze major role in social communication has been of interest to many researchers (see [5, 43, 44] for reviews). In particular, literature explains how eye-gaze is likely to be used to perceive and understand the emotional and mental states of others and subsequently how it may be a reliable source to anticipate their actions. Thus, rather than gaze-cue not being able to direct attention to a place other than the signalled location, participants may attribute a specific intention to the eye-gaze by retaining their attention specifically at the inferred-at location or the signalled placeholder and not to the entire hemifield or nearby placeholders. Consequently, if these social mechanisms are involved in the specific attentional orienting triggered by eye-gaze, it seems logical to expect that when the spatial cue is non-biologically relevant as an arrow, such a mechanism would not be activated, and attention would be rather spread to nearby objects or the other extreme of the object when larger objects are used [18] following perceptual grouping laws.

Therefore, in order to investigate social attention, paradigms that measure qualitative rather than quantitative differences between biologically and non-biologically relevant stimuli should be used, since the standard gaze-cueing paradigm has proven not to be suitable to capture differences in the attentional orienting effect elicited by social and non-social cues [27]. It will be interesting for future research to explore whether the aforementioned qualitative differences between eye-gaze and arrow cues can be observed in populations with reduced social abilities, such as people with autism spectrum disorder or schizophrenia. Perhaps, in these populations, no difference between social and non-social attentional cues may be observed.

## Author Contributions

**Conceptualization:** Jeanette A. Chacón-Candia, Juan Lupiáñez, Andrea Marotta.

**Investigation:** Jeanette A. Chacón-Candia.

**Supervision:** Juan Lupiáñez, Andrea Marotta.

**Writing – original draft:** Jeanette A. Chacón-Candia.

**Writing – review & editing:** Jeanette A. Chacón-Candia, Juan Lupiáñez, Maria Casagrande, Andrea Marotta.

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
