## [Decision Letter · Decision Letter 0]

20 Apr 2022

PONE-D-22-03696Eye-Gaze direction triggers a more specific attentional orienting compared to arrowsPLOS ONE

Dear Dr. Chacón Candia,

Thank you for submitting your manuscript to PLOS ONE. After careful consideration, we feel that it has merit but does not fully meet PLOS ONE’s publication criteria as it currently stands. Therefore, we invite you to submit a revised version of the manuscript that addresses the points raised during the review process.

Based on the comments of two reviewers I am inviting you to resubmit with ‘major revisions’. As you will see from their reports, both reviewers have particular concerns about the statistical analyses, with Reviewer 2 highlighting the need to address sampling and statistical power. Reviewer 1 expresses additional concerns about the intelligibility of the manuscript as presented. As these are standard criteria for acceptance, I encourage you to address all comments thoroughly. Reviewer 2 has made a number of suggestions as to other research on attentional cueing you might consider in rewriting, that will likely broaden the reach of your research. Again I would encourage you to consider your findings in light of the boarder perspective.

We look forward to receiving your revised manuscript.

Kind regards,

Nuala Brady

Academic Editor

PLOS ONE

Journal Requirements:

(This work was supported by a research project (PID2020-114790GB-I00) by the Spanish Ministry of Science and Innovation/AEI (https://www.ciencia.gob.es/) to JL,  a research project (B-SEJ-572-UGR20) by the Regional Government of Andalusia (https://www.juntadeandalucia.es/) to AM and a Ph.D. fellowship in Psychology and Cognitive Science by Sapienza the University of Rome (https://www.uniroma1.it) to JCH-C.)

Please include your amended Funding Statement within your cover letter. We will change the online submission form on your behalf."

Reviewers' comments:

Reviewer's Responses to Questions

**Comments to the Author**

1. Is the manuscript technically sound, and do the data support the conclusions?

Reviewer #1: Partly

Reviewer #2: Yes

2. Has the statistical analysis been performed appropriately and rigorously? 

Reviewer #1: No

Reviewer #2: Yes

3. Have the authors made all data underlying the findings in their manuscript fully available?

Reviewer #1: Yes

Reviewer #2: Yes

4. Is the manuscript presented in an intelligible fashion and written in standard English?

Reviewer #1: No

Reviewer #2: Yes

5. Review Comments to the Author

Reviewer #1: I was a little confused by some of your statistical analyses. It sounds like you did separate analyses for general cueing effects and hemifield effects, for each of the placeholder-present and placeholder-absent conditions. Why were these conditions analyzed separately? It seems like the conclusion that shifts of attention to gaze and arrow cues differ in the specificity of their attentional focus (which is modulated by the placeholders) necessitates directly testing these interactions. For example, in Experiment 1, you really have 4 cue-target relations (same location, opposite location, same hemifield, opposite hemifield), 2 cue types (gaze, arrow), and 2 placeholder conditions (present, absent), which could/should be analyzed in one ANOVA. There are some interesting patterns in the data that are not being addressed by analyzing these separately (e.g., in Exp. 2, for gaze cues the opposite-location/opposite-group RTs are faster than both same group and opposite group, and for arrow cues the opposite location is faster than the opposite group, both of which are unexpected).

It appears that eye movements were not monitored (understandable, given the shift to online data collection), but is it possible that there are differences in eye movements to the two cue types that could explain the patterns of results? Do gaze cues elicit more overt orienting, which could account for the greater specificity of the attentional allocation?

I have a minor concern about the cues themselves and how that might affect the observed pattern. For the arrow cues, the horizontal line is present and then the arrowhead appears at cue onset, whereas for the gaze cues the pupils are present and then move to the left or the right at cue onset (which may be programmed as the appearance of a new object, but I imagine to the participant is perceived as the movement of the pupils). How does appearance of a new object vs. movement of an existing object affect cueing and the orienting of attention? I.e., is the difference observed really between gaze and arrows, or is it because of lower-level differences in object processing?

The final paragraph at the end of the conclusion either needs to be removed or expanded to include citations and a more thorough discussion of why understanding different mechanisms for gaze and arrow cues is important. Right now it is too vague and doesn’t add anything to the paper.

Although not a major concern, to fully understand the differences between these tasks it might be useful to dig into the data in Exp. 2 and 3 further:

1) Were there distance effects in Exp. 2 or 3? In both these experiments the cued/opposite placeholder is on the horizontal meridian and then the two within-group placeholders are at a graded distance above or below this location. It appears that both locations within the group were combined together for analysis, but one would imagine a differential “spread” of attention to nearby and more distant placeholders. It might be useful to directly assess this.

2)Were there differences between visual fields (upper/lower and left/right)? When one group is in the lower VF, the opposite group is always in upper VF (and vice versa). There might be asymmetries in how attention is allocated (i.e., easier to shift attention downward than upward, might see less of a group effect when attention is directed upward and then more easily shifts downward on invalidly cued trials). Similarly, you might see effects of left vs. right (easier to shift attention from left to right), or an interaction between left/right and up/down, such that effects when cued group is in upper left will be very different when it is in lower right.

There are some typographical and grammatical errors throughout the manuscript. These are generally minor and don’t impede understanding of the paper, but a thorough proof-reading is recommended. Here are a few examples:

p. 2, ln. 23 – remove ‘the’ before others

p. 2, ln. 34 – missing ‘e’ on Kingstone

p. 6, ln. 117 – sited should be seated

p. 7, ln. 154 – remove ‘so call’

Reviewer #2: The study includes three experiments that examined whether the inclusion of a placeholder would modulate orienting of attention in a Posner cueing paradigm to socially relevant versus socially non -relevant directional cues. Experiment 2 and 3 were conducted online and Experiment 1 was an in-person laboratory experiment. Results support previous findings that have shown that when an object is present eye-gaze cueing facilitates attention to object part whereas arrows to the entire object.

Overall, the manuscript is well presented. I suggest the authors redraft the introduction to include a more comprehensive yet description of relevant research in the field ( see recommendations below). More importantly is the issue of sample size selection, composition and justification ( or lack thereof). I have some reservations about the the joined analysis of exp 2 and 3 which I detail below

Review Comments to the Author

Introduction:

The comparison of biologically/socially relevant attentional cues (e.g eye-gaze, pointed hands, head orientation and body orientation) versus non-social cues ( e.g arrows) has a rich history in the field of attention and more generally social cognition and perception. The introduction would benefit from a more in-depth description of some of the research in the field. Below are some references to relevant papers that the authors may wish to include. These are papers that have compared attentional orienting to social vs. non-social directional cues as well as papers that have examined perceptual representation of these directional cues. And more broadly, paper that have discuss how individual differences may account for the results we see in social attention with respect to cue type.

• Capozzi, F., & Ristic, J. (2018). How attention gates social interactions. Annals of the New York Academy of Sciences, 1426(1), 179-198.

• Dalmaso, M., Castelli, L., & Galfano, G. (2020). Social modulators of gaze-mediated orienting of attention: A review. Psychonomic Bulletin & Review, 27(5), 833-855.

• Guzzon, D., Brignani, D., Miniussi, C., & Marzi, C. A. (2010). Orienting of attention with eye and arrow cues and the effect of overtraining. Acta Psychologica, 134(3), 353-362.

• Cooney SM, O’Shea A, Brady N (2015) Point Me in the Right Direction: Same and Cross Category Visual Aftereffects to Directional Cues. PLoS ONE 10(10): e0141411.

Sample size, Composition & Statistical Power

How was the sample size estimated for the three experiments? 37 experiment 1,75 experiment 2, 26 for experiment 3. Justification for sample size needs to be given.

The majority of the sample in all three experiments is female. Please comment on this as a limitation with reference to previous research that has identified sex differences in spatial orienting of attention in similar Posner cueing experiments know differences in how males and female orient their attention (see Cooney, Brady & Ryan 2017; Bayliss, di Pellegrino & Tipper, 2005, Mitsuda, T., Otani, M., & Sugimoto, S. (2019).

Exp 3 is a replication but with less than half the sample of experiment. The author’s then go on to run a combined analysis for experiment 2 & 3. Further justification is required here. Presumably, the combined analysis was conducted because Exp 3 contained a very small sample size. However, it’s important to note that the participants in Exp 3 only took part in one condition. While the authors mention this choice they do not justify it – why did the participants only do the placeholder present condition? The authors need to justify this choice in relation to the power of the sample in Exp 3. In general, the power and justification for sample size in all 3 experiments requires justification and discussion.

I have reservations about the combined analysis as it does not follow best research practice. Given that the result of the cue-type ( arrow, eye gaze) * cue-target relation (same-location/same-hemifield vs. opposite-location/opposite- hemifield) changes (i.e. it becomes statistically significant) when the sample size is increased in the combined analysis and the study does not seem to be pre-registered, I recommend removing the combined analysis. Or at the very least detailing the choice and the possible limitations of this approach.

Method: for the arrow trials a horizontal line was placed…presumably this was a horizontal line with an arrowhead – this need to be specified.

Experiment 1 - Method

Please report the size and dimensions of the computer screen/monitor.

How was ‘unpredictability’ of the target X and O operationalized? Was there an equal amount of trials for X and O targets?

SOA: Would the authors expect different results if the SOA was manipulated?

Why was 300ms chosen?

Instead of describing three studies refer to each as Experiment 1, Experiment 2, Experiment 3.

Exp 2 and 3 are online experiments whereas Exp 1 was in-person. As Exp 2 & 3 were online, considerably less control over the way the stimuli were viewed., i.e did all participant take part on laptop/Pc’s?

Post-hocs should be reported for significant interactions. The authors refer to figures to make inferences about the interactions – at the very least the post- hocs should be in the supplementary.

There are several grammatical and spelling errors

For example - Page 2 line 39. Sentence beginning 'They found is.. '

should read 'They found that'

Line 117 'sited' should read 'seated'

6. PLOS authors have the option to publish the peer review history of their article (what does this mean?). If published, this will include your full peer review and any attached files.

Reviewer #1: No

Reviewer #2: No

---

## [Author Response · Author response to Decision Letter 0]

13 Jul 2022

I hereby enclose the revised version of manuscript number PONE-D-22-03696R1 titled “Eye-Gaze direction triggers a more specific attentional orienting compared to arrows”. 

We very much appreciate your comments as well as the reviewers’ suggestions; we have tried to address all of them in this new version of the manuscript. We agree that the points raised by you and by the reviewers have contributed to strengthening the document. Below we detail the changes in the manuscript in response to the reviewers’ comments.

We hope that after these final changes the manuscript is suitable for publication. Furthermore, we are willing to make any new change if requested.

Thank you again for the opportunity, and for your time and consideration.

Sincerely,

J CH-C (on behalf of the other authors)

Reviewer #1

I was a little confused by some of your statistical analyses. It sounds like you did separate analyses for general cueing effects and hemifield effects, for each of the placeholder-present and placeholder-absent conditions. Why were these conditions analyzed separately?. It seems like the conclusion that shifts of attention to gaze and arrow cues differ in the specificity of their attentional focus (which is modulated by the placeholders) necessitates directly testing these interactions. For example, in Experiment1, you really have 4 cue-target relations (same location, opposite location, same hemifield, opposite hemifield), 2 cue types (gaze, arrow), and 2 placeholder conditions (present, absent), which could/should be analyzed in one ANOVA. There are some interesting patterns in the data that are not being addressed by analyzing these separately (e.g., in Exp. 2, for gaze cues the opposite-location/opposite-group RTs are faster than both same group and opposite group, and for arrow cues the opposite location is faster than the opposite group, both of which are unexpected).

Response: Thank you for your comments. The analyses you suggested are now reported in the revised version of the paper (See Experiment 1 & 2, Results section).

It appears that eye movements were not monitored (understandable, given the shift to online data collection), but is it possible that there are differences in eye movements to the two cue types that could explain the patterns of results? Do gaze cues elicit more overt orienting, which could account for the greater specificity of the attentional allocation?

Response: Several studies have suggested that there are no differences between the overt attention produced by eye-gaze and arrow stimuli (Khun & Benson, 2007; Khun et al., 2010; Khun & Kingstone, 2009). However, even if such a difference between the two cue types does exist, we would expect it to affect in the same manner in both, the placeholder-present and the placeholder-absent conditions. Nevertheless, our patter of results shows a difference between the two stimuli, but only when placeholder objects are presented in the scene, suggesting that the observed differences are not due to the over attention produced by eye-gaze and arrows, but may be due to the distinct ways in which people select objects in response to them.

References

Kuhn, G., & Benson, V. (2007). The influence of eye-gaze and arrow pointing distractor cues on voluntary eye movements. Perception & psychophysics, 69(6), 966-971.

Kuhn, G., Benson, V., Fletcher-Watson, S., Kovshoff, H., McCormick, C. A., Kirkby, J., & Leekam, S. R. (2010). Eye movements affirm: automatic overt gaze and arrow cueing for typical adults and adults with autism spectrum disorder. Experimental Brain Research, 201(2), 155-165.

Kuhn, G., & Kingstone, A. (2009). Look away! Eyes and arrows engage oculomotor responses automatically. Attention, Perception, & Psychophysics, 71(2), 314-327.

I have a minor concern about the cues themselves and how that might affect the observed pattern. For the arrow cues, the horizontal line is present and then the arrowhead appears at cue onset, whereas for the gaze cues the pupils are present and then move to the left or the right at cue onset (which may be programmed as the appearance of a new object, but I imagine to the participant is perceived as the movement of the pupils). How does appearance of a new object vs. movement of an existing object affect cueing and the orienting of attention? I.e., is the difference observed really between gaze and arrows, or is it because of lower-level differences in object processing?

Response: This is an interesting point that we have now commented on as a limitation in the revised version of the paper (see, General discussion section). Nonetheless, if this phenomenon influenced orienting of attention, the same influence should be observed in the placeholder-absent condition. However, no gaze-arrow differences exist when no placeholders are presented on the scene.

The final paragraph at the end of the conclusion either needs to be removed or expanded to include citations and a more thorough discussion of why understanding different mechanisms for gaze and arrow cues is important. Right now, it is too vague and doesn’t add anything to the paper.

Response: Thank you for the observation. This point has been re-address in the revised version of the paper.

Although not a major concern, to fully understand the differences between these tasks it might be useful to dig into the data in Exp. 2 and 3 further:

1) Were there distance effects in Exp. 2 or 3? In both these experiments the cued/opposite placeholder is on the horizontal meridian and then the two within-group placeholders are at a graded distance above or below this location. It appears that both locations within the group were combined together for analysis, but one would imagine a differential “spread” of attention to nearby and more distant placeholders. It might be useful to directly assess this.

Response: Thank you for your suggestion; this is an interesting issue. Although we did not include this analysis in the revised version of the paper, we explored, as you suggested, the data related to the cue-target distance when the placeholder-present grouping effect was to be analysed. By doing this analysis in experiment 2, we indeed found an effect of distance (near vs fare; F1,74=36.76, p=<.001,η2p=0.332), showing that, in general, reaction times were faster when targets appeared at the grouping-near location than on the grouping-far location. The interaction cue-type X validity X distance (F1,74=5.38, p=.023,η2p=0.068) was also significant, revealing that when arrows were used as a cue, beside a main effect of validity (F1,74=13.09, p=<.001,η2p=0.150; lower reaction times for valid [M=559, SD=80.34] than for invalid [M=571, SD=82.13] trials), participants responded faster when the target appeared at the near than at the fare location in both the valid (M=548 SD=76.24 vs M=570 SD=88.86) and invalid trials (M=566 SD=85.27 vs M=575 SD=85.25). Importantly, no interaction of validity X distance was found when analysing the arrow-grouping effect (p>.05). When eye gaze was used, the distance effect was also significant (F1,74=19.16, p=<.001,η2p=0.206), showing again that participants responded faster when the target appeared at the nearest than at the furthest location in both the valid (M=558 SD=82.73 vs M=569 SD=80.40) and invalid trials (M=556 SD=72.97 vs M=575 SD=82.73). However, nor the validity nor the interaction of validity X distance were significant (p>.05).

Experiment 3 has been removed in the revised version of the paper, which is the reason why we are not reporting the suggested analysis from that experiment.

2) Were there differences between visual fields (upper/lower and left/right)? When one group is in the lower VF, the opposite group is always in upper VF (and vice versa). There might be asymmetries in how attention is allocated (i.e., easier to shift attention downward than upward, might see less of a group effect when attention is directed upward and then more easily shifts downward on invalidly cued trials). Similarly, you might see effects of left vs. right (easier to shift attention from left to right), or an interaction between left/right and up/down, such that effects when cued group is in upper left will be very different when it is in lower right.

Response: Thank you for your suggestion; this is an interesting issue. Although we did not include this analysis in the revised version of the paper, as it was not the main goal of the study, we explored, as you suggested, the data related to the hemifields up/down and left/right.

First, the analysis reported a main effect of hemifield (up vs down) when an arrow was presented as a cue (F1,74=8.94, p=.004,η2p=0.108), showing that in this placeholder-present condition, participants responded faster when targets appeared at the upper than at the lower hemifield in both valid (M=567, SD=91.72 vs M=573, SD=92.74) and invalid trials (M=566, SD=82.67 vs M=584, SD=94.37). When eye gaze was used, no up/down hemifield effect was found (p>.05). Secondly, when analysing the left/right hemifield effect, no hemifield or any related interactions were found (all p>.05).

Experiment 3 has been removed in the revised version of the paper, which is the reason why we are not reporting the suggested analysis from that experiment.

There are some typographical and grammatical errors throughout the manuscript. These are generally minor and don’t impede understanding of the paper, but a thorough proof-reading is recommended.

Response: Thank you for the observation. The signalled errors have been corrected in the revised version of the paper. We also have reviewed the entire manuscript again to try to point out and correct any other possible errors.

Reviewer #2

The study includes three experiments that examined whether the inclusion of a placeholder would modulate orienting of attention in a Posner cueing paradigm to socially relevant versus socially non –relevant directional cues. Experiment 2 and 3 were conducted online and Experiment 1 was an in-person laboratory experiment. Results support previous findings that have shown that when an object is present eye-gaze cueing facilitates attention to object part whereas arrows to the entire object.

Overall, the manuscript is well presented. I suggest the authors redraft the introduction to include a more comprehensive yet description of relevant research in the field (see recommendations below). More importantly is the issue of sample size selection, composition and justification (or lack thereof). I have some reservations about the joined analysis of exp 2 and 3 which I detail below.

Introduction:

The comparison of biologically/socially relevant attentional cues (e.g eye-gaze, pointed hands, head orientation and body orientation) versus non-social cues (e.g arrows) has a rich history in the field of attention and more generally social cognition and perception. The introduction would benefit from a more in-depth description of some of the research in the field. Below are some references to relevant papers that the authors may wish to include. These are papers that have compared attentional orienting to social vs. non-social directional cues as well as papers that have examined perceptual representation of these directional cues. And more broadly, paper that have discuss how individual differences may account for the results we see in social attention with respect to cue type.

• Capozzi, F., & Ristic, J. (2018). How attention gates social interactions. Annals of the New York Academy of Sciences, 1426(1), 179-198.

• Dalmaso, M., Castelli, L., & Galfano, G. (2020). Social modulators of gaze-mediated orienting of attention: A review. Psychonomic Bulletin & Review, 

27(5), 833-855.

• Guzzon, D., Brignani, D., Miniussi, C., & Marzi, C. A. (2010). Orienting of attention with eye and arrow cues and the effect of overtraining. Acta Psychologica, 134(3), 353-362.

• Cooney SM, O’Shea A, Brady N (2015) Point Me in the Right Direction: Same and Cross Category Visual Aftereffects to Directional Cues. PLoS ONE 10(10): e0141411.

Response: Thank you for your suggestions. We have now considered the studies you have mentioned for the revised version of the paper. 

Sample size, Composition & Statistical Power

How was the sample size estimated for the three experiments? 37 experiment 1, 75 experiment 2, 26 for experiment 3. Justification for sample size needs to be given.

Response: There was no experiment of reference for our first experiment, as this was the first time our paradigm was used. We could use as reference the study by Wiese et al., (2013), but they did not compare arrows and gaze, which was critical for our experiment. Instead, we could use Marotta et al. (2012) experiments, in which objects instead of group of objects (i.e., placeholders) were used, but they did compare gaze with arrow cues. Marotta et al. (2012) used samples of 24 and 30 participants, so we decided to use a sample of 36 (37 at the end) participants for Experiment 1. For experiment 2, we duplicated the sample size because of the online modality. We agree that in experiment 3, the sample was not very big. We decide to include this experiment because was basically a replication, and could be useful for performing a joint analysis with the placeholders data from Experiment 2. However, we agree that this experiment alone was not sufficiently powered and have decided to remove it from the revised version of the paper. 

Unfortunately, as we did not have a clear reference experiment, we did not perform a priori power analysis. However, a sensitivity analysis carried out now for Experiment 1, with a sample size of 37 participants, with � = .05 and a power of 1-�=.90, showed that the experiment had enough sensitivity as to detect a minimum effect size of η2p=0.0202 (f= 0.143); the same analysis carried out for Experiment 2 (75 participants) showed that the experiment was sufficiently power as to detect a minimum effect size of η2p=0.0100 (f= 0.100). Importantly, these minimum effect size that could be detected by our design were smaller than the observed effect sizes for the critical interactions. 

References

Marotta, A., Lupianez, J., Martella, D., & Casagrande, M. (2012). Eye gaze versus arrows as spatial cues: Two qualitatively different modes of attentional selection. J Exp Psychol Hum Percept Perform, 38(2), 326-335. https://doi.org/2011-12265-001 [pii] 10.1037/a0023959

Wiese, E., Zwickel, J., & Müller, H. J. (2013). The importance of context information for the spatial specificity of gaze cueing. Attention, Perception, & Psychophysics, 75(5), 967–982. doi: 10.3758/s13414-013-0444-y

The majority of the sample in all three experiments is female. Please comment on this as a limitation with reference to previous research that has identified sex differences in spatial orienting of attention in similar Posner cueing experiments know differences in how males and female orient their attention (see Cooney, Brady & Ryan 2017; Bayliss, di Pellegrino &Tipper, 2005, Mitsuda, T., Otani, M., & Sugimoto, S. (2019).

Response: In fact, this is one of the main goals of a study we have now in preparation. We have already collected the full sample (female: 93, male: 90) and analysed the data. Results show that neither the interaction sex X cue-type (F1,182=0.421, p=.517), sex X validity (F3,546=0.644, p=.423), nor the sex X cue-type X validity (F3,546=1.086, p=.299) was significant. This lead us to the conclusion that no differences between males and females seem to be present in our study when using this task to dissociate between social and non-social attention.

Exp 3 is a replication but with less than half the sample of experiment. The author’s then go on to run a combined analysis for experiment 2 & 3. Further justification is required here.

Response: Experiment 3 was added in order to replicate the findings. However, we understand that the sample size was notably smaller than Exp 2, the reason why following your advice, we have removed experiment 3 and the combined analysis from the revised version of the paper. Furthermore, we are carrying out one study (in preparation) in which we have used this task (placeholders present condition only) to compare social vs non-social attention in male and female students from STEM vs social sciences/humanities careers. Therefore, we are now confident that the observed pattern of data for the placeholders condition in Exp 2 replicates. Indeed, we replicated the Cue-Type x Validity, F3,546= 13.85, p<.001. Again, t-test analyses show that the general cueing effect was significant for both arrows (p<.001) and gaze (p=.015), but the object-based effect was only significant for arrows (p<.001), not for gaze (p=.322).

Presumably, the combined analysis was conducted because Exp 3 contained a very small sample size. However, it’s important to note that the participants in Exp 3 only took part in one condition. While the authors mention this choice they do not justify it – why did the participants only do the placeholder present condition? The authors need to justify this choice in relation to the power of the sample in Exp 3. In general, the power and justification for sample size in all 3 experiments requires justification and discussion.

Response: We have removed Exp 3 and the combined analysis in the revised version of the paper.

I have reservations about the combined analysis as it does not follow best research practice. Given that the result of the cue-type (arrow, eye gaze) * cue-target relation (same-location/same-hemifield vs. opposite-location/opposite- hemifield) changes (i.e. it becomes statistically significant) when the sample size is increased in the combined analysis and the study does not seem to be pre-registered, I recommend removing the combined analysis. Or at the very least detailing the choice and the possible limitations of this approach.

Response: We have removed Exp 3 and the combined analysis in the revised version of the paper.

Method

For the arrow trials a horizontal line was placed… presumably this was a horizontal line with an arrowhead – this need to be specified.

Response: Thank you for the observation. This has been specified in the revised version of the paper (See Experiment 1, Method section: “Apparatus and stimuli”).

Experiment 1 – Method: Please report the size and dimensions of the computer screen/monitor.

Response: Thank you for the observation. This has been mentioned in the revised version of the paper (See Experiment 1, Method section: “Apparatus and stimuli”).

How was ‘unpredictability’ of the target X and O operationalized? Was there an equal amount of trials for X and O targets?

Response: Thank you for the observation. This has been specified in the revised version of the paper (See Experiment 1, Method section: “Procedure”).

SOA: Would the authors expect different results if the SOA was manipulated? Why was 300ms chosen?

Response: The reason why an SOA of 300 ms was chosen was that recent meta-analytic evidence (Chacón-Candia et al., 2022 [under review]) has shown that SOAs of 200 to 400 ms report a stronger magnitude of gaze and arrow cueing effects (Figure 1), therefore, in order to shorten the duration of the experiment and because it is not relevant to the issues addressed in this article, we decided only to use an overall mean duration reported in previous evidence.

Figure 1 (see on the Response to reviewers file)

Decrease of the standardized cueing effect with SOA. Filled circles represent studies using cues with long duration, whereas empty circles represent short cues (≤ 300 ms of duration). The size of the circles was proportional to the number of participants contributing to the effect. Dashed and dotted lines show the decrease in the cueing effects with long and short cue displays, respectively.

References

Chacón-Candia J A, Román-Caballero R, Aranda-Martín B, Lupiáñez J, Casagrande M, Marotta A. (2022). No quantitative differences between eye-gaze and arrow cues: A meta-analytic answer and a call for qualitative differences. [Manuscript submitted for publication].

Instead of describing three studies refer to each as Experiment 1, Experiment 2, Experiment 3.

Response: Thank you for the observation. This issue has been re-addressed in the revised version of the paper. 

Exp 2 and 3 are online experiments whereas Exp 1 was in-person. As Exp 2 & 3 were online, considerably less control over the way the stimuli were viewed, i.e did all participant take part on laptop/Pc’s?

Response: Yes, we made sure to make it sufficiently clear in the instructions given to participants that it was necessary to use a PC/laptop to successfully complete the task (responses to the target were to be made by pressing a key on the keyboard).

Post-hocs should be reported for significant interactions. The authors refer to figures to make inferences about the interactions – at the very least the post- hocs should be in the supplementary.

Response: All relevant comparisons are now reported in the revised version of the paper.

There are several grammatical and spelling errors.

Response: Thank you for the observation. The signalled errors have been corrected in the revised version of the paper. We also have reviewed the entire manuscript again to try to point out and correct any other possible errors.

---

## [Editor Report · Decision Letter 1]

26 Jul 2022

PONE-D-22-03696R1Eye-Gaze direction triggers a more specific attentional orienting compared to arrowsPLOS ONE

Dear Dr. Chacón Candia,

Thank you for submitting your manuscript to PLOS ONE. After careful consideration, we feel that it has merit but does not fully meet PLOS ONE’s publication criteria as it currently stands. Therefore, we invite you to submit a revised version of the manuscript that addresses the points raised during the review process. Thank you for replying to the reviewers’ comments. The MS has clearly benefited from these revisions and should have more widespread appeal given your consideration of the boarder literature on attentional cueing by socially relevant stimuli. However, as editor, I have difficulties discerning whether you have addressed the concerns of both reviewers adequately, both of whom were chosen as experts in the area. Before considering the MS for publication in PLOS ONE, I would like to see the following issues addressed. The first is general, the others are specific to Reviewer 1 and 2 respectively 1.
In many instances you simply reply to the reviewers’ concerns personally but do not incorporate these issues into the revised MS. While this may be appropriate for some concerns (e.g., if a reviewer raises an issue that is not part of your design it may be appropriate to communicate privately, as it were), in many cases the issues flagged by the reviewers are ones that readers will be interested in and have identical questions to the reviewers. As an obvious one, the shift from lab based to online work brings questions as to the size of the device a participant might use (PC screen, tablet, phone) which is potentially very important to grouping and proximity – while you reply to the reviewer you do not seem to have incorporated this into the revised MS. Please provide a clean copy of the revised MS which (a) excludes removed material (i.e. all the crossed out text), (b) uses track changes to highlight revised and new material and (c) please in the reply to reviewers indicate exactly where in the revised MS the changes have been made (pXX, lines XX-XX on revised MS), and (d) where you are replying privately to a reviewer and are NOT making changes to the MS, please say so explicitly  2.
Reviewer 1 raises the important issue of how the statistical analysis was preformed, recommending for Exp 1 that you start with a repeated measures ANOVA with factors of cue-type (2 levels), placeholder-condition (2 levels) and validity (4 levels) to analyse the RT data. You have now done this but it very hard to see whether the ‘separate t-tests’ that you are refer to follow directly from this main ANOVA? In presenting the results for this ANOVA you make no reference to cue-type – I assume this is because there is ‘not significant’ but as this is the crux of the research, it needs brief mention. The same issue arises for Exp 2(a)
Please clarify that these analyses were post-hoc tests or planned comparisons arising from the main ANOVA. Please submit an ANOVA table for Exp 1 and Exp 2. If you do not wish to include in the MS that is fine, but it would be helpful for the evaluation of the MS to have ANOVA results to hand(b)
The language used is very confusing, e.g., by a ‘general cueing effect’ do you mean a cueing effect that is independent of the cue type (arrow/gaze)? If so, please state this. You seem to have taken the Reviewer’s advice as to how to analyse the data but are sticking with the terminology from the first MS?(c)
It would be helpful if you would present the statistical results with direct reference to Figure 3 (for Exp 1) and Figure 6 (for Exp 2), and use an asterisk to denote significance in the plots where this occurs. Both these figures should have a visible and labelled y-axis 3.
Reviewer 2 highlights the essential issue of statistical power and sample size. For Exp 1 you justify your use of sample size with reference to published work. This is not good practice as much published work in experimental psychology is likely under-sampled. (a)
Please provide a retrospective power analysis for Exp 1 as suggested by the reviewer, using conventional choices of effect size, power etc. As this is a repeated measures design you may wish to consider https://psycnet.apa.org/record/2019-45517-001(b)
Please address how your sample size compares with that recommended by the power analysis (c)
Please do the same for Exp 2. Here the sample size is considerable greater than for Exp 1 with no justification except for the ease of recruiting online. (d)
Although you have removed the formal comparison between Exp 1 and Exp 2 as recommended by the reviewer, you still need to consider the implications of different sample sizes for your conclusions, e.g., is it appropriate to conclude that gaze acts differently than arrows but only in the case of grouped stimuli? Because you do see a 3-way interaction in Exp 2 but not in Exp 1? Is it possible that the difference in the results come down to sample size? Given the importance of power analyses, sample size etc and the considerable issues of moving from laboratory based to online research across the two experiments, it may be wise to discuss with your research team conducting Exp 2 in the lab with formal power analysis? Minor issues1.
With regard to reference 28 (Chacón-Candia J A, Román-Caballero R, Aranda-Martín B, Lupiáñez J, Casagrande M, Marotta A.. please use the standard method of citing online publications and do not use “Manuscript submitted for publication”: You can find the correct citation at https://psyarxiv.com/2.
Table 1 You refer to percent error yet use abbreviation IR (I assume this is ‘incorrect response’?). Please use either one or the other as it is confusing when the abbreviation does not match the long text.

We look forward to receiving your revised manuscript.

Kind regards,

Nuala Brady

Academic Editor

PLOS ONE
---

## [Author Response · Author response to Decision Letter 1]

20 Oct 2022

1. Editor

Dear Dr. Chacón Candia,

Thank you for submitting your manuscript to PLOS ONE. After careful consideration, we feel that it has merit but does not fully meet PLOS ONE’s publication criteria as it currently stands. Therefore, we invite you to submit a revised version of the manuscript that addresses the points raised during the review process.

Thank you for replying to the reviewers’ comments. The MS has clearly benefited from these revisions and should have more widespread appeal given your consideration of the boarder literature on attentional cueing by socially relevant stimuli. 

However, as editor, I have difficulties discerning whether you have addressed the concerns of both reviewers adequately, both of whom were chosen as experts in the area. Before considering the MS for publication in PLOS ONE, I would like to see the following issues addressed. The first is general, the others are specific to Reviewer 1 and 2 respectively

In many instances you simply reply to the reviewers’ concerns personally but do not incorporate these issues into the revised MS. While this may be appropriate for some concerns (e.g., if a reviewer raises an issue that is not part of your design it may be appropriate to communicate privately, as it were), in many cases the issues flagged by the reviewers are ones that readers will be interested in and have identical questions to the reviewers. 

Response: Thank you for your comments. We have tried to address this general suggestion in the revised version of the paper.

* As an obvious one, the shift from lab based to online work brings questions as to the size of the device a participant might use (PC screen, tablet, phone) which is potentially very important to grouping and proximity – while you reply to the reviewer you do not seem to have incorporated this into the revised MS. 

Response: This issue has been explained more clearly in the revised version of the manuscript in the “Procedure” section of experiment 2 (p.15; lines 329-335 on revised MS).

Please provide a clean copy of the revised MS which: 

(a) excludes removed material (i.e. all the crossed out text) 

Response: We have considered this in the new version of the marked-up copy of the manuscript.

(b) uses track changes to highlight revised and new material 

Response: We have considered this in the new version of the marked-up copy of the manuscript.

(c) please in the reply to reviewers indicate exactly where in the revised MS the changes have been made (pXX, lines XX-XX on revised MS).

Response: We have considered this in the new version of the marked-up copy of the manuscript.

(d) where you are replying privately to a reviewer and are NOT making changes to the MS, please say so explicitly 

Response: We have considered this in the new version of the marked-up copy of the manuscript.

Reviewer 1

* Reviewer 1 raises the important issue of how the statistical analysis was preformed, recommending for Exp 1 that you start with a repeated measures ANOVA with factors of cue-type (2 levels), placeholder-condition (2 levels) and validity (4 levels) to analyse the RT data. You have now done this but it very hard to see whether the ‘separate t-tests’ that you are refer to follow directly from this main ANOVA?

Response: Yes, t-tests are done following the interaction to analyse the general cueing and hemifield effects specifically. This has now been specified in the revised version of the paper in the "Design" section of experiment 1 (p.9; lines 202-204 on revised MS). We hope it is now clearer. 

* In presenting the results for this ANOVA you make no reference to cue-type – I assume this is because there is ‘not significant’ but as this is the crux of the research, it needs brief mention. The same issue arises for Exp 2

Response: Thank you for your observation; this has now been specified in the revised version of the paper in the “Results” section for both experiment 1 (p.12; lines 253-254 on revised MS) and experiment 2 (p.18; line 400-401 on revised MS).

(a) Please clarify that these analyses were post-hoc tests or planned comparisons arising from the main ANOVA. 

Response: This is now specified in the new version of the paper (p.9; lines 202-204 on revised MS). T-tests are planned comparisons always comparing the two relevant conditions for our effects of interest: general cueing and hemifield/grouping effects. When we have no specific a priori hypotheses, we performed partial ANOVAs following the significant interactions. We hope this is now clearer in the manuscript.

* Please submit an ANOVA table for Exp 1 and Exp 2. If you do not wish to include in the MS that is fine, but it would be helpful for the evaluation of the MS to have ANOVA results to hand

Response: Below you will find the complete tables of the results of the ANOVA analyses for experiments 1 and 2, respectively. We believe it is not necessary to include these tables in the manuscript.

Experiment 1.

(You can find the image at the attached document called "Response to Reviewers").

Experiment 2.

(You can find the image at the attached document called "Response to Reviewers"). 

(b) The language used is very confusing, e.g., by a ‘general cueing effect’ do you mean a cueing effect that is independent of the cue type (arrow/gaze)? If so, please state this. 

Response: We are sorry for the confusion. The “general-cueing effect” refers to the comparisons between same-location/same-hemifield and opposite-location/opposite-hemifield conditions (p.9; line 204-206 on revised MS), and the hemifield-effect to that between same-hemifield and opposite-hemifield conditions (p.9; line 206-207 on revised MS). Depending on whether the validity effect is modulated by Cue type (Exp 2) or not (Exp1), t-tests for these effects are performed specifically for each cue-type or independently of it.

In other words, the general cueing effect is the standard effect observed when the target appears only in the placeholders located on each side of the cue. Although we understand the confusion with this term, we have used it following previous similar research (Marotta et al., 2012).

We hope this is clearer now in the manuscript. 

Reference

Marotta, A., Lupiánez, J., Martella, D., & Casagrande, M. (2012). Eye gaze versus arrows as spatial cues: two qualitatively different modes of attentional selection. Journal of Experimental Psychology: Human Perception and Performance, 38(2), 326. doi: 10.1037/a0023959.

• You seem to have taken the Reviewer’s advice as to how to analyse the data but are sticking with the terminology from the first MS?

Response: Indeed, we have followed his/her advice regarding the analyses. However, for analysing the interactions of interest (regarding the variable validity), we have kept the terminology general-cueing and hemifield/grouping effects as we believe these terms are helpful to differentiate distinct types of attentional orienting. Nevertheless, we hope that the revised version of the manuscript clarifies how these effects were analysed.

(c) It would be helpful if you would present the statistical results with direct reference to Figure 3 (for Exp 1) and Figure 6 (for Exp 2), and use an asterisk to denote significance in the plots where this occurs. Both these figures should have a visible and labelled y-axis

Response: Thank you for pointing this out. We have now included the references of figures 3 and 6 in the description of the results of experiment 1 (p.12; line 251 on revised MS) and experiment 2 (p.19; line 416-417 on revised MS) on the revised version of the manuscript. We also added asterisks to the figures to denote significance in the plots where this occurs, and we have labelled the y-axis.

Reviewer 2 

* Reviewer 2 highlights the essential issue of statistical power and sample size. For Exp 1 you justify your use of sample size with reference to published work. This is not good practice as much published work in experimental psychology is likely under-sampled. 

Response: We have merged in one answer the critical points highlighted by Reviewer 2. You can find the complete response at the end of this comments section. 

(a) Please provide a retrospective power analysis for Exp 1 as suggested by the reviewer, using conventional choices of effect size, power etc. As this is a repeated measures design you may wish to consider https://psycnet.apa.org/record/2019-45517-001

(b) Please address how your sample size compares with that recommended by the power analysis 

(c) Please do the same for Exp 2. Here the sample size is considerable greater than for Exp 1 with no justification except for the ease of recruiting online. 

(d) Although you have removed the formal comparison between Exp 1 and Exp 2 as recommended by the reviewer, you still need to consider the implications of different sample sizes for your conclusions, e.g., is it appropriate to conclude that gaze acts differently than arrows but only in the case of grouped stimuli?. Because you do see a 3-way interaction in Exp 2 but not in Exp 1? Is it possible that the difference in the results come down to sample size?. 

Given the importance of power analyses, sample size etc and the considerable issues of moving from laboratory based to online research across the two experiments, it may be wise to discuss with your research team conducting Exp 2 in the lab with formal power analysis?

Response: We agree with the reviewer that it is not good practice to base sample size estimation on previous estimations. Indeed, we currently conduct a priori power analysis to estimate the needed sample size for our studies. However, the data from the experiments reported in the current paper were collected before this practice was fully established in our lab. As far as we know, the best is rather to report sensitivity analysis, showing the minimum effect size that could be detected with our sample size. Therefore, we now report sensitivity analyses for both Experiments 1 and 2 (p's.6 and 7; lines 139-143 and p.14; lines 309-313, respectively, on revised MS). 

Of course, we agree that replicating in the lab with an a priori estimated sample size would be ideal. However, we have already replicated online the same pattern of data observed in Experiment 2 (placeholders present condition), in which we investigated in a large sample size of 200 men and women students from different careers (STEM vs social sciences) the effects observed for arrows and gaze. Although no differences between gender or sex were observed, the overall pattern was replicated: whereas arrows led to both general-cueing and grouping effects gaze only produced a significant general-cueing effect. We are therefore confident that the data pattern is reliable and replicable online. 

Minor issues

1. With regard to reference 28 (Chacón-Candia J A, Román-Caballero R, Aranda-Martín B, Lupiáñez J, Casagrande M, Marotta A. please use the standard method of citing online publications and do not use “Manuscript submitted for publication”: You can find the correct citation at https://psyarxiv.com/

Response: Thank you for pointing this out. The citation has been corrected in the revised version of the manuscript (p.28; lines 632-635 on revised MS). 

2. Table 1 You refer to percent error yet use abbreviation IR (I assume this is ‘incorrect response’?). Please use either one or the other as it is confusing when the abbreviation does not match the long text.

Response: Thank you for your observation; this issue has been corrected in both table 1 (p.10; line 219 on revised MS) and table 2 (p.17; lines: 378 on revised MS) of the revised version of the manuscript.

---

## [Decision Letter · Decision Letter 2]

12 Jan 2023

Eye-Gaze direction triggers a more specific attentional orienting compared to arrows

PONE-D-22-03696R2

Dear Dr. Chacón Candia,

We’re pleased to inform you that your manuscript has been judged scientifically suitable for publication and will be formally accepted for publication once it meets all outstanding technical requirements.

Kind regards,

Avid Roman-Gonzalez, Ph.D.

Academic Editor

PLOS ONE

Additional Editor Comments (optional):

Reviewers' comments:

Reviewer's Responses to Questions

**Comments to the Author**

1. If the authors have adequately addressed your comments raised in a previous round of review and you feel that this manuscript is now acceptable for publication, you may indicate that here to bypass the “Comments to the Author” section, enter your conflict of interest statement in the “Confidential to Editor” section, and submit your "Accept" recommendation.

Reviewer #3: (No Response)

Reviewer #4: (No Response)

2. Is the manuscript technically sound, and do the data support the conclusions?

Reviewer #3: Partly

Reviewer #4: Partly

3. Has the statistical analysis been performed appropriately and rigorously? 

Reviewer #3: Yes

Reviewer #4: Yes

4. Have the authors made all data underlying the findings in their manuscript fully available?

Reviewer #3: No

Reviewer #4: Yes

5. Is the manuscript presented in an intelligible fashion and written in standard English?

Reviewer #3: Yes

Reviewer #4: (No Response)

6. Review Comments to the Author

Reviewer #3: In the light of your method that addresses attentional shifts triggered by these different types of stimuli and trial sequence. Results are interesting and my main question is (1) whether your study may work just in trail sequence or comparing with Markov chains. Please can you clarify the following observations before giving a final feedback?

I. INTRODUCTION

(2) About RELATED WORK, authors develop methods based on "trial sequence", I believe they should introduce that term, of course related to "orienting of attention" and "eye-tracking", there are few articles.

II- METHODS

(3) Authors have compared gaze with arrow cues. They have 288 trials for each block, but they have not clearly stated by sequence is analized

III. EXPERIMENTS

(4) How many trials were used for experiment 2?

IV. GENERAL DISCUSSION.

(5) Authors have discussed about P1 & N1 and N2 & P300 by Marotta and colleagues, but my understanding of their experiment is a SOA or CTOA around 1000 ms, therefore is not clearly related to arrows at SOA or CTOA of 300 ms. I have looked briefly at Schoolar Google and some articles appeared

https://scholar.google.com/scholar?hl=en&as_sdt=0%2C5&q=%22orienting+of+attention%22+%22300+ms%22+p300+ctoa&btnG= , authors should address which of these articles can be added to your discussion.

(6) Can you explain in detail your SOA or CTOA of 300 ms in the context of << social directional cues like gaze might produce an effect of a

501 similar nature, as the common effect observed with the standard gaze cueing paradigm

502 and the general-cueing effect observed in our experiments, they must produce an extra

503 effect that restricts attention to the specifically looked-at location >>

MINOUR ISSUES

(7) Standardize your writing-up, e.g. "cuetype" --- "cue-type", "300ms" --- "300 ms"

(8) Data availability, I can't find a link or a similar one.

Reviewer #4: The paper does not contain the item conclusions, it only contains the item general discussion. It would be convenient to add it

7. PLOS authors have the option to publish the peer review history of their article (what does this mean?). If published, this will include your full peer review and any attached files.

Reviewer #3: No

Reviewer #4: No

---

## [Editor Report · Acceptance letter]

16 Jan 2023

PONE-D-22-03696R2 

Eye-Gaze direction triggers a more specific attentional orienting compared to arrows 

Dear Dr. Chacón Candia:

I'm pleased to inform you that your manuscript has been deemed suitable for publication in PLOS ONE. Congratulations! Your manuscript is now with our production department. 

Kind regards, 

on behalf of

Professor Avid Roman-Gonzalez 

Academic Editor

PLOS ONE